# A Multistep Prediction Model for the Vibration Trends of Hydroelectric Generator Units Based on Variational Mode Decomposition and Stochastic Configuration Networks

**DOI:** 10.3390/s23249762

**Published:** 2023-12-11

**Authors:** Shaokai Yan, Fei Chen, Jiandong Yang, Zhigao Zhao

**Affiliations:** State Key Laboratory of Water Resources Engineering and Management, Wuhan University, Wuhan 430072, China; shaokai.yan@whu.edu.cn (S.Y.); chenfei@whu.edu.cn (F.C.)

**Keywords:** hydroelectric generator unit, vibration trend prediction, variational mode decomposition, stochastic configuration networks, recursive multistep prediction model

## Abstract

Accurately predicting the changes in turbine vibration trends is a key part of the operational condition maintenance of hydropower units, which is of great significance for improving both the operational condition and operational efficiency of hydropower plants. In this paper, we propose a multistep prediction model for the vibration trend of a hydropower unit. This model is based on the theoretical principles of signal processing and machine learning, incorporating variational mode decomposition (VMD), stochastic configuration networks (SCNs), and the recursive strategy. Firstly, in view of the severe fluctuations of the vibration signal of the unit, this paper decomposes the unit vibration data into intrinsic mode function (IMF) components of different frequencies by VMD, which effectively alleviates the instability of the vibration trend. Secondly, an SCN model is used to predict different IMF components. Then, the predicted values of all the IMF components are superimposed to form the prediction results. Finally, according to the recursive strategy, a multistep prediction model of the HGU’s vibration trends is constructed by adding new input variables to the prediction results. This model is applied to the prediction of vibration data from different components of a unit, and the experimental results show that the proposed multistep prediction model can accurately predict the vibration trend of the unit. The proposed multistep prediction model of the vibration trends of hydropower units is of great significance in guiding power plants to adjust their control strategies to reach optimal operating efficiency.

## 1. Introduction

As the proportion of renewable energy sources like wind and solar grows within the power production system, addressing the challenges of grid integration for these intermittent energy sources has become crucial [1,2]. Hydroelectric power, being the pioneer in power generation technologies, serves as a linchpin for peak shifting and frequency regulation, and it often complements these intermittent energy sources, ensuring the grid operates smoothly and stably [3,4,5]. Given this backdrop, it is imperative for hydroelectric units to possess a broad operational range to cater to the regulatory demands of the system. Yet their expansive operational scope and intricate structure result in a heightened risk of turbine failures, and traditional methods of planned and shutdown maintenance no longer suffice for the safe operation of hydropower plants, making condition-based maintenance using historical operation data the prevailing trend [6].

In recent years, a significant proportion of the research concerning the operational state of hydraulic turbines has been centered on fault diagnosis. Utilizing data from vibration, pressure pulsation, and other unit-specific metrics to ascertain the operating condition of turbines has emerged as a focal area of investigation, yielding a series of notable findings [7,8,9,10]. For example, Zheng et al. [8] proposed a method for identifying the flow state in the leafless zone of the draft tube based on empirical wavelet transform (EWT) and adaptive convolutional neural networks (ACNNs), which were used to analyze and identify the different operating states of the pump water wheel. Dao et al. [11] collected fault signals caused by unit sediment abrasion through acoustic vibration sensors on the turbine runner and determined the fault frequency of the signals through wavelet transform (WT), ensemble empirical mode decomposition (EEMD), and fast Fourier transform (FFT) methods, which is an important guide for the operation and maintenance of turbines which have been maintained over sediments. Therefore, developing a high-performance HGU vibration trends prediction model to accomplish accurate forecasts of HGU state trends will help power plant decision-makers build more appropriate operation and maintenance plans. These methods have a certain reference value in detecting and identifying abnormal states in hydropower units, but it is worth noting that these data-driven methods essentially belong to the category of ex post facto analysis and can only play a role in the event of equipment failure. Meanwhile, the health state of HGUs will show a trend of gradual deterioration due to the influence of the operating environment and equipment life, which will lead to the above fault diagnosis methods being unable to detect early equipment failures in time [12].

During real-world power station inspections, personnel often prioritize monitoring fluctuations in hydraulic turbine vibration and oscillation signals. If these operational metrics exceed specific thresholds, alarms are activated. The emphasis on vibration and oscillation signals as key detection indices stems from two primary reasons: first, these signals are exceptionally convenient to monitor, and second, the majority of hydraulic turbine malfunctions manifest externally as excessive vibration or oscillation [13,14,15,16]. Consequently, accurately forecasting the trend changes in hydraulic turbine vibration and oscillation has emerged as a novel research focus [17,18,19,20,21,22]. For instance, Bi et al. [18] proposed a fused metrics method based on Pearson and distance correlation to select suitable condition variables and integrated the BLSTM network to establish a prediction model that accurately forecast the vibration trend in hydraulic turbines. Similarly, Zhou et al. [21] amalgamated signal processing, feature selection, and prediction modules within a multi-objective optimization algorithm, resulting in an accurate and efficient model for predicting the vibration trends of hydropower units.

The above data-driven methods have achieved good application results in the field of vibration trends prediction of HGUs. However, most of these models only involve single-step prediction and cannot predict the change of vibration trend of hydroelectric units for a long time in the future. How to construct a multistep prediction model and realize multistep prediction of unit vibration trend is the focus of this paper. In general, the multistep prediction model of vibration trend is mainly constructed based on five multistep prediction strategies: Recursive [23], Direct [24], DirRec [25], Multiple-Input–Multiple-Output (MIMO) [26], and DIRMO [27]. Among them, the recursive strategy has the advantages of simplicity and easy operation. Meanwhile, the strategy adopts the method of rolling prediction to build a multistep prediction model step by step, which is in line with the idea of HGU vibration trends prediction. For this reason, this paper constructs a multistep model for HGU vibration trends prediction by the recursive strategy.

Essentially, HGU vibration trends prediction is a kind of time series prediction based on HGUs’ historical data. For the time series prediction, Fu et al. [13] generalized the vibration trends prediction of HGUs into two aspects: signal processing and data regression. Among them, recursive signal processing methods such as EMD [28,29], EEMD [30,31] and LMD are widely used. However, recursive methods like EMD have problems such as insufficient theory, serious modal aliasing, and inability to select modal components. For EEMD, it generates redundant IMFs during the decomposition process, which may not have practical significance or contribute to the analysis. This can increase the complexity of signal decomposition and add unnecessary computations. To address these problems, Dragomiretski et al. [32] proposed variational mode decomposition (VMD). Unlike recursive decomposition algorithms, such as EMD, VMD searches for variational models by iterative methods and decomposes the signal into IMF components of different frequency bands. In this article, VMD is chosen as a preprocessing method for HGU signals to alleviate the nonstationarity problem of the acquired vibration signals.

In recent years, with the continuous development of machine learning and deep learning, AI models are gradually replacing traditional statistical models. For example, Fu et al. [13] proposed an HGU vibration trends prediction model based on OVMD and LSSVM, which overcame the disadvantage of the poor prediction accuracy of a single model. Xiong et al. [19] introduced a deep learning framework into the field of HGU vibration trends prediction, constructed a hybrid CNN-LSTM prediction model, and realized the effective prediction of HGU vibration trends. However, all of the above models suffer from the problem of difficult parameter settings, which seriously reduces the generalization ability of the models. A stochastic configuration network (SCN) [33], as a kind of stochastic parameter neural network, employs a model optimizer with low computational cost and high efficiency in the randomization algorithm. Therefore, it has good performance in model efficiency and accuracy. In this article, an SCN model is utilized to predict the IMF components of the VMD, and the predicted values of all IMF components are superimposed in the final prediction results.

For fast and accurate multistep prediction of vibration trends in hydroelectric units, we have conducted the following studies and made the following contributions:(1)We propose a hybrid VMD and SCN-based vibration trend prediction model for hydroelectric units.(2)We introduce the recursive strategy to enhance the VMD–SCN model, resulting in a multistep prediction model for the vibration trend of hydroelectric units.(3)We apply the proposed multistep prediction model to the trend prediction of two different signal types: vibration signals and swing signals. This further validates the effectiveness and practicality of the proposed method.

The remainder of the paper is as follows: Section 2 introduces the methods and principles of the multistep prediction model. Section 3 shows the concrete flow of the proposed framework. Section 4 carries out case study research on a real Francis turbine. Finally, Section 6 obtains the conclusions of the study.

## 2. Methods and Principles

### 2.1. Variational Mode Decomposition (VMD)

Aiming to solve the problems of empirical mode decomposition (EMD), such as endpoint effect, mode aliasing, and insufficient theoretical basis, Dragomiretski and Zosso [32] proposed a new adaptive signal processing method named VMD. Different from recursive decomposition algorithms such as EMD, VMD searches variational modes through iterative methods and decomposes signals into IMF components of different frequency bands. VMD has good applications in fault diagnosis [34,35], signal noise reduction [36] and parameter identification [37]. The specific calculation method of VMD is as follows:

(1)Taking the minimum sum of the estimated bandwidths of each modal component as the objective function, the constrained variational problem obtained is:(1)$$\begin{array}{c} \min\left\{\sum_{k=1}^{K}{\left\|\partial_{t}\left[\left(\delta \left(t\right)+\frac{j}{\pi t}\right)*u_{k}(t)\right]e^{-j\omega_{k}t}\right\|}_{2}^{2}\right\}\\ s.t\;\sum_{k=1}^{n}u_{k}\left(t\right)=x(t) \end{array}$$
where K represents the number of intrinsic mode function (IMF) components, $\partial_{t}$ is the first-order partial derivative of the function with respect to time, $\delta \left(t\right)$ is the unit impulse function, ∗ is the convolution symbol, $u_{k}\left(t\right)$ represents the k-th IMF component, $\omega_{k}$ represents the center frequency of the k-th mode component, and $x\left(t\right)$ is the original signal.(2)To simplify the calculation, the quadratic penalty factor and Lagrange multiplier are introduced to transform Equation (1) into an unconstrained problem.
(2)$$\begin{array}{c} L\left(\left\{u_{k}\right\},\left\{\omega_{k}\right\},\lambda \right)=\alpha \sum_{k=1}^{K}{\left\|\partial_{t}\left[\left(\delta \left(t\right)+\frac{j}{\pi t}\right)*u_{k}(t)\right]e^{-j\omega_{k}t}\right\|}_{2}^{2}+\\ \left\|x(t)-\sum_{k=1}^{K}u_{k}(t)\right\|+\langle \lambda (t),x(t)-\sum_{k=1}^{K}u_{k}(t)\rangle \end{array}$$
where α is the quadratic penalty factor, and $\lambda \left(t\right)$ is the Lagrange multiplier.(3)According to the alternating direction multiplier method to find the saddle point of Equation (2), the specific process is as follows:

Set the number of mode decomposition K, initialize the frequency domain ${\hat{u}}_{k}^{1}$, the center frequency $\omega_{k}^{1}$, and the Lagrange multiplier ${\hat{\lambda }}^{1}$, and then calculate the IMF $u_{k}$ and center frequency $\omega_{k}$ according to Equation (3).
(3)$$\begin{array}{l} {\hat{u}}_{k}^{n+1}=\frac{\hat{x}\left(\omega \right)-\sum_{i\neq k}{\hat{u}}_{i}\left(\omega \right)+\frac{\hat{\lambda }\left(\omega \right)}{2}}{1+2\alpha {\left(\omega -\omega_{k}\right)}^{2}}\\ \omega_{k}^{n+1}=\frac{\int_{0}^{\infty }\omega \left|{\hat{u}}_{k}\left(\omega \right)\right|d\omega }{\int_{0}^{\infty }\left|{\hat{u}}_{k}\left(\omega \right)\right|d\omega } \end{array}$$

Calculate the Lagrange multiplier λ according to Equation (4):(4)$${\hat{\lambda }}^{n+1}\left(\omega \right)={\hat{\lambda }}^{n}\left(\omega \right)+\zeta_{1}\left(\hat{x}\left(\omega \right)-\sum {\hat{u}}_{k}^{n+1}\left(\omega \right)\right)$$
where $\zeta_{1}$ is the noise tolerance.

$u_{k}$, $\omega_{k}$ and λ are iteratively updated successively until the criterion in Equation (5) is satisfied.
(5)$$\sum_{k}\frac{{\left\|u_{k}^{n+1}-u_{k}^{2}\right\|}_{2}^{2}}{{\left\|u_{k}^{2}\right\|}_{2}^{2}}<E$$
where E is the discriminant accuracy.

### 2.2. Stochastic Configuration Networks (SCNs)

As a new type of randomly weighted neural network with a supervisory mechanism, SCNs are different from conventional feedforward neural networks. SCNs are gradually constructed according to the supervisory mechanism, which restricts the random input weights and the specific value range of biases. This supervisory mechanism guarantees the general approximation property of the SCN model generated by a given nonlinear mapping. SCNs are widely used in energy consumption prediction [38,39], industrial production [40,41] and state recognition [42]. The detailed process of the SCN model is described as follows:

(1)Given a training data set W,P∈ℝδ+d×n, where W represents input data, P represents output data, δ is the dimension of input data, d is the dimension of output data, and n is the number of samples. Assuming that the SCN model has S−1 hidden nodes, the output ZS−1 of the SCN is:(6)ZS−1=∑i=1S−1βigi(ωiTW+bi)S=2,3,4,⋯,Z0=0
where βi=[βi,1,βi,2,⋅⋅⋅,βi,d]T is the output weight vector of the i-th hidden node, ωi and bi are the input weight vector and bias of the i-th hidden node. g⋅ represents an activation function of the SCN model. Meanwhile, the current residual eS−1 can be calculated according to Equation (7):(7)eS−1=Z−ZS−1=[eS−1,1,eS−1,2,⋯,eS−1,d](2)The SCN introduces a supervisory mechanism to assign parameters to hidden nodes. The specific supervisory mechanism forms are as follows:(8)gS=gSωSTw1+bS,gSωSTw2+bS,⋯,gSωSTwn+bSTeS−1,j,gS2≥bg2(1−r−μS)eS−1,j2,j=1,2,⋯,d
where eS−1,j,gS is the inner product of the vector eS−1,j and gS, gS is the output of the S-th hidden node, for ∀g∈Γ (Γ denotes a spanned function space), 0≤g≤bg, bg∈ℝ+, r is the regularization parameter ranging from 0 to 1. μS is a sequence of nonnegative real numbers with limS→∞μS=0 and 0<μS≤1−r, and the hidden node optimal parameters ωs and bs are determined according to the supervisory mechanism.(3)Use the least squares method to calculate the hidden layer output weights:(9)[β1,β2,⋯,βS]=argminZ−∑j=1Sβjgj2

Continue to increase the hidden nodes, and repeat Equation (6) to Equation (8) until the model residual $\left\|e_{S}\right\|$ reaches the expected error tolerance χ or the number of hidden nodes reaches the maximum number of hidden nodes $S_{\max}$, and the optimal model is finally output. To facilitate the understanding of the SCN model, the basic structure diagram of the SCN model is shown in Figure 1.

### 2.3. Recursive Multistep Prediction Strategy

Constructing a multistep prediction model of the HGU’s vibration trends is of great significance to maintain the safe and stable operation of HGUs. This paper uses the sliding window concept to construct a recursive multistep prediction model. As shown in Figure 2, the recursive multistep prediction model is essentially based on the recursive use of the single-step model, and the rolling prediction of the model is realized by continuously adding the predicted values to the training model. For example, given a historical data set ${\left[X\left(1\right),X\left(2\right),\cdots \cdots ,X(q)\right]}^{T}$ of vibration signals with equal time step length, the 2-step prediction model uses the prediction results of the single-step prediction model to reconstruct the training set sample, and predicts the vibration trend value of the unit at the $\left(q+2\right)$-th moment according to the single-step prediction value $Y\left(q+1\right)$ plus the last $\left(q-1\right)$ values in the set. By analogy, the 5-step prediction model uses the prediction results $Y\left(q+1\right)$,$Y\left(q+2\right)$,$Y\left(q+3\right)$, and $Y\left(q+4\right)$ of the 4-step prediction model plus the last $\left(q-4\right)$ values in the set to predict the vibration signal value at the $\left(q+5\right)$-th moment. In this paper, q is set as 20, and a 5-step prediction model of the vibration tendencies of HGUs is constructed by using a recursive multistep prediction strategy.

### 2.4. Model Prediction Performance Evaluation Indicator

In this paper, four indexes, including root mean square error (*RMSE*), mean square error (*MSE*), mean absolute error (*MAE*), and mean absolute percentage error (*MAPE*), are introduced to evaluate the prediction performance of the model. The calculation formulas are as follows:(10)$$\begin{array}{c} RMSE=\sqrt{\frac{1}{m_{1}}\sum_{i=1}^{m_{1}}{\left({\hat{y}}_{i}-y_{i}\right)}^{2}}\;\;\;MSE=\frac{1}{m_{1}}\sum_{i=1}^{m_{1}}{\left({\hat{y}}_{i}-y_{i}\right)}^{2}\\ MAE=\frac{1}{m_{1}}\sum_{i=1}^{m_{1}}\left|{\hat{y}}_{i}-y_{i}\right|\;\;\;MAPE=\frac{100\%}{m_{1}}\sum_{i=1}^{m_{1}}\left|\frac{{\hat{y}}_{i}-y_{i}}{y_{i}}\right| \end{array}$$
where $m_{1}$ is the total number of sample points, $\hat{y}$ is the predicted value, and y is the true value.

## 3. Process of Vibration Trend Prediction Method for HGU Based on VMD and SCN

The flow chart of the vibration trends prediction for the HGU based on VMD and the SCN model is shown in Figure 3. The steps of the method used in this paper can be summarized as follows:(1)Data acquisition and storage. The vibration signals from the hydraulic turbine are detected by vibration sensors and swing sensors installed at the power station. The online monitoring system collects, records, and stores long-term operational data of the hydropower unit using the data acquisition system and storage server for data collection, display, and storage.(2)Signal decomposition. Clean and screen the unit state operation data and divide them into training set and test set according to a certain ratio. Then the VMD algorithm discussed in Equations (1)–(5) is utilized to decompose the samples of the test set and the training set to obtain the IMF components of different frequencies.(3)Single-step vibration trend prediction. At first, the input and output data of different IMF components of the samples of the training set are obtained using the form of sliding window. Then, train the SCN model using all the IMF components of the training set (see Equations (6)–(9) for the specific algorithmic process) and apply them to the corresponding IMF components of the test set. Finally, the predicted values of all IMF components are summarized to obtain the single-step prediction results of the vibration trend of the hydropower unit.(4)Multistep vibration trend prediction. In accordance with the recursive multistep prediction strategy in Section 2.3, this paper adopts the rolling prediction method to construct a multistep prediction model to realize the five-step forward prediction of the vibration trend of the unit, and measures the performance of the prediction model by the prediction performance evaluation index mentioned in Section 2.4.

## 4. Single-Step Prediction Experiment of Vibration Tendency for HGU Based on VMD and SCN

In this section, the single-step prediction research of the vibration trends for the HGU based on VMD and the SCN model is carried out. The VMD algorithm contains two important parameters, the number of IMF components K and the penalty factor α. In this paper, the penalty factor α is set as 2000, and the center frequency method is used to determine the optimal decomposition level K of VMD. Meanwhile, the scale factor $\lambda_{1}$ of the input weight and bias of the SCN model is set as {0.5, 1, 5, 10, 30, 50, 100, 150, 200, 250}, the regularization parameter r is set as {0.9, 0.99, 0.9999, 0.99999, 0.999999}, the error tolerance χ is set as 0.01, and the maximum number of hidden nodes $S_{\max}$ is set as 30.

### 4.1. Experimental Data Description

The experimental data in this paper are from the No. 1 unit of a hydropower station in China, whose unit model is LJ267-LJ-175. In this experiment, vibration signals in the *X*-direction of the turbine upper frame and swing signals in the *X*-direction of the turbine guide bearing are taken as the research objects. The monitoring data of the unit from 1 August 2021 to 31 August 2021, are selected and the monitoring interval of the data is 1 h. The specific situation of the unit is shown in Figure 4.

The monitoring data of the unit are processed through data cleaning methods such as deleting null fragments, deleting abnormal values, and interpolation, and 600 sets of *X*-direction vibration data of the turbine upper frame and *X*-direction swing data of the turbine guide bearing are finally obtained. The specific waveforms are shown in Figure 5. As shown in Figure 5, the vibration signals of the turbine upper frame and the swing signals of the turbine guide bearing show strong nonlinear variation trends, which brings certain challenges to the accurate prediction of the unit vibration trends. Meanwhile, the data are divided into a training set and a test set according to a ratio of 2:1.

### 4.2. Prediction Results Analysis

According to Section 2.1, VMD is used to decompose the unit vibration signals into IMF components of different frequencies. To eliminate the influence of mode aliasing on the prediction results, the center frequency method is used to determine the optimal decomposition level *K.* In this paper, the IMF central frequency under different decomposition levels *K* is analyzed and refers to the mode aliasing determination method; that is, when the central frequencies of three adjacent IMF components are on the same order of magnitude, mode aliasing occurs. Taking the *X*-direction vibration signal of the turbine upper frame as an example, it is found from Table 1 that when *K* is 8, the center frequencies of the three adjacent IMF components are all on the order of 0.2, so the optimal decomposition level of the vibration signal is 7. According to the same method, the optimal decomposition level of the swing data of the turbine guide bearing in the *X*-direction is eight.

In this paper, VMD is used to decompose the training set and test set of vibration signals into IMF components of different frequencies, and the input and output values of the training set and test set are constructed based on these IMF components. Then, SCN models of different IMF components are trained through the training set samples, and the test set samples of different IMF components are predicted through the trained SCN models. The prediction results of different IMF components are shown in Figure 6 and Figure 7. From Figure 6 and Figure 7, it can be seen that there are no significant deviations between the predicted results of the SCN model and the actual values for the different IMF components, and the overall trend of predicted and actual values is consistent, which shows that the SCN model has good predictive performance.

The predicted values of the different IMF components are superimposed to obtain the final prediction results. As shown in Figure 8, the VMD–SCN model performs well in predicting the vibration signals of two different parts of the unit. Meanwhile, the error diagram and prediction correlation analysis diagram are introduced to evaluate the prediction results. The results show that the VMD–SCN model has an error of less than 3 μm in predicting the turbine upper frame vibration signal, with a coefficient of determination (R-squared) of 98.1% between the actual and predicted values. Similarly, for the turbine guide bearing swing signal, the prediction error is less than 5 μm, and the R-squared between the actual and predicted values reaches 98.5%.

### 4.3. Comparative Experiment

To verify the superiority of the VMD–SCN model in the single-step prediction of the vibration trends of the HGU, the VMD–DBN, VMD–BPNN, BPNN, SCN, DBN, EEMD–SCN, and EEMD–DBN models are introduced for comparative experiments. The single-step prediction research of the vibration trends of the HGU is carried out by using the above models. The prediction results of the above models on the vibration signals in the *X*-direction of the turbine upper frame are shown in Figure 9, and the prediction results of the models on the first and last day are also given. It can be seen from Figure 9 that the results predicted by the VMD–SCN model are closer to the actual values compared with other models, which preliminarily verifies that the VMD–SCN model has good predictive performance.

Meanwhile, to quantitatively analyze the prediction results of different models, RMSE, MAE, MAPE, and MSE indicators are introduced to evaluate the predictive performance of the models. The specific results are shown in Figure 10. As can be seen from Figure 10, compared with the other models (VMD–DBN, VMD–BPNN, BPNN, SCN, DBN, EEMD–SCN, and EEMD–DBN), the RMSE indicator of the VMD–SCN model decreased by 1.071, 1.709, 3.455, 2.119, 2.341, 0.861 and 1.094, respectively. The MAE indicator decreased by 0.890, 1.422, 2.817, 1.731, 1.870, 0.724, and 0.850, respectively. The MAPE indicator decreased by 0.615, 0.977, 1.930, 1.178, 1.278, 0.504, and 0.588, respectively, and the MSE indicator decreased by 2.997, 5.870, 17.902, 8.146, 9.521, 2.228, and 3.086, respectively. Based on the above results, the VMD–SCN model achieves the best predictive performance for the *X*-direction vibration signals of the turbine upper frame.

Similarly, the prediction results of the VMD–SCN and other models on the swing signals in the *X*-direction of the turbine guide bearing are shown in Figure 11, and the prediction results of the models on the first and last day are also shown. It can be seen from Figure 11 that the results predicted by the VMD–SCN model are closer to the actual value compared with other models, which indicates that the VMD–SCN model is also applicable to the prediction of the swing signal trends of HGUs.

The prediction performance indicators of different models are provided in Figure 12. As can be seen from Figure 12, compared with other models (VMD–DBN, VMD–BPNN, BPNN, SCN, DBN, EEMD–SCN, and EEMD–DBN), the RMSE indicator of the VMD–SCN model decreased by 2.182, 6.393, 5.513, 2.441, 4.853, 0.915, and 3.056, respectively; the MAE indicator decreased by 1.730, 4.856, 4.200, 2.095, 4.151, 0.763, and 2.566, respectively; the MAPE indicator decreased by 1.892, 5.330, 4.492, 2.296, 4.565, 0.862, and 2.793, respectively; and the MSE indicator decreased by 11.187, 59.692, 46.623, 13.146, 37.838, 3.533, and 18.337, respectively. Based on the above results, the VMD–SCN model still achieves the best predictive performance on the *X*-direction swing signals of the turbine guide bearing.

Through the analysis of the prediction results of the two sets of data, it is effectively verified that the VMD–SCN model has great potential in the single-step prediction of vibration signals of HGUs.

## 5. Multistep Prediction Experiment on the Vibration Tendency of HGU Based on VMD and SCN

The single-step prediction experiment in Section 4 effectively verifies that the VMD–SCN model has good single-step predictive performance, but single-step prediction has limited practical utility. The multistep prediction of unit vibration trends can provide more support for maintaining the safe and stable operation of units. According to the multistep prediction strategy described in Section 2.3, a recursive multistep prediction model is constructed in this paper, and multistep prediction experiments on the vibration trends of HGUs based on VMD and the SCN model are carried out. All experimental data are consistent with the *X*-direction vibration data of the turbine upper frame and the *X*-direction swing data in Section 4. Meanwhile, all comparative experiments are kept consistent with those in Section 4.

The multistep prediction (two-step to five-step prediction) results of the different models on the vibration signals in the *X*-direction of the turbine upper frame are shown in Figure 13. As can be seen from Figure 13, with the increase of the number of prediction steps, the models’ prediction errors continue to accumulate, resulting in different degrees of degradation in the predictive performance of the models. Single models such as the BPNN, DBN, and SCN models, show poor predictive performance in multistep prediction, with significant fluctuations occurring during the prediction process leading to decreased prediction accuracy. The prediction accuracy of VMD–DBN, EEMD–DBN, and other combined models is improved compared with the single model, but the prediction results of these models have significant errors in some time periods. Moreover, this phenomenon becomes more obvious as the number of predicted steps increases. Based on the prediction results of all the models, it can be seen that the multistep prediction results of the VMD–SCN model are closer to the actual values than the other models.

Similar to Section 4, to quantitatively analyze the prediction results of different models, RMSE, MAE, MAPE, and MSE indicators are used to evaluate the multistep prediction performance of the models. The specific prediction results are shown in Figure 14. As can be seen from Figure 14, the performance indicators (RMSE, MAE, MAPE, and MSE) of the two-step prediction model of the VMD-SCN are 1.248, 1.024, 0.706, and 1.557; the performance indicators of the three-step prediction model reach 1.782, 1.439, 0.990, and 3.176; the performance indicators of the four-step prediction model reach 2.527, 2.003, 1.378, and 6.388; and the performance indicators of the five-step prediction model reach 3.301, 2.630, 1.813, and 10.899. Compared with other models (VMD–DBN, VMD–BPNN, BPNN, SCN, DBN, EEMD–SCN, and EEMD–DBN), the VMD–SCN model has the most effective multistep predictive performance.

The multistep prediction (two-step to five-step prediction) results of different models for the swing signals in the *X*-direction of the turbine guide bearing are shown in Figure 15. Similar to the previous analysis, it can be concluded from Figure 15 that with the increase of the number of prediction steps the models’ prediction errors continue to accumulate, resulting in different degrees of degradation in the prediction performance of the models. Comparing the prediction results of the combined model with the single model, it is also verified that the combined model can improve the prediction performance of the model to a certain degree. However, the VMD–SCN model has some deviation in the four-step and five-step predictions, with errors reaching up to 20 μm in certain time periods. Nevertheless, the VMD–SCN model can capture the overall trends of the vibration signals and demonstrates good overall prediction performance. Combining all the prediction results, it can be concluded that the multistep prediction results of the VMD–SCN model are closer to the actual values compared with the other models.

Similarly, the multistep prediction performance indicators of the different models are shown in Figure 16. As can be seen from Figure 16, the two-step prediction model performance indicators (RMSE, MAE1, MAPE, and MSE1) of the VMD–SCN are 2.024, 1.542, 1.697, and 4.095; the performance indicators of the three-step prediction model reach 3.577, 2.850, 3.116, and 12.793; the performance indicators of the four-step prediction model reach 5.129, 4.013, 4.412, and 26.309; and the performance indicators of the five-step prediction model reach 7.436, 5.801, 6.403, and 55.697. Compared to the other models (VMD–DBN, VMD–BPNN, BPNN, SCN, DBN, EEMD–SCN, and EEMD–DBN), the VMD–SCN model has the most effective performance in multistep prediction.

Based on the above two sets of multistep prediction experiments and comparing the prediction performances of the different models, the following conclusions can be drawn:(1)As the number of prediction steps increases, the RMSE, MSE, and other prediction indicators of the different models show an upward trend. This indicates that with the accumulation of prediction errors, the prediction performance of the models will decrease. It indirectly verifies that the prediction range of multistep prediction is limited. The reference value of a model’s prediction results becomes small when the prediction time exceeds a certain range.(2)Comparing the prediction indicators of combined models such as the VMD–SCN and single models such as the SCN, it is observed that the prediction indicators of single models are higher, indicating lower prediction accuracy. Moreover, the prediction accuracy of single models significantly decreases with an increase in the number of prediction steps. To a certain extent, combined models overcome the limitation of low prediction accuracy in single models and hold significant importance in predicting the vibration trends of HGUs.(3)From the prediction performance of the VMD–SCN and EEMD–SCN models, it can be observed that the model using the VMD method of decomposition achieves better prediction results. Similarly, comparing the prediction evaluation indicators of the VMD–DBN and EEMD–DBN models, it is found that the prediction performance of the VMD–DBN is stronger than that of the EEMD–DBN model. Therefore, the VMD method adopted in this paper is more suitable for the analysis of unit vibration data.(4)Comparing the prediction results of the VMD–SCN model with the other seven models, it is found that the VMD–SCN model shows the best prediction performance on both datasets, which strongly verifies that the VMD–SCN model has a great multistep prediction performance of HGU vibration trends.

## 6. Conclusions

To accurately predict the vibration trends of HGUs, this paper proposes a combined multistep prediction model based on VMD and an SCN model. Firstly, considering the significant fluctuations in the vibration signals of HGUs, VMD is used to decompose the vibration signals into IMF components of different frequencies, which effectively alleviates the instability of vibration signal fluctuations. Secondly, the SCN model is used to predict the different IMF components separately. Then, the predicted values of the different IMF components are superimposed to form the prediction results. Finally, according to the recursive strategy, the prediction results are input as new input values, achieving multistep prediction of the vibration trends in HGUs. The VMD–SCN model is applied to two different types of datasets of the upper frame vibration signals and the turbine guide bearing swing signals of a unit, and seven models are introduced for comparative experiments. The main conclusions are as follows:(1)To a certain extent, the hybrid model overcomes the shortcomings of the single models and their low prediction accuracy, which is of great significance for the prediction of unit vibration trends.(2)Comparing the prediction results of the VMD–-SCN, VMD–DBN, EEMD–SCN, and EEMD–DBN, it is verified that using the VMD decomposition method is more suitable for the analysis of unit vibration data.(3)By comparing the prediction results of the VMD–SCN model with the other seven models, it is found that the VMD–SCN has the best prediction effect among all the models, showing strong prediction performance, which is helpful for assisting the decision makers at the power plant to formulate more reasonable operation and maintenance strategies.

Although the proposed method has achieved notable prediction results, there are still some pressing issues that require urgent investigation: (1) The operating state of the hydropower unit undergoes a gradual deterioration process, and it is possible that the known training samples and the unknown test samples may not belong to the same distribution. Therefore, the introduction of transfer learning theory into trend prediction has become one of our future research directions. (2) This paper solely addresses trend prediction research. Our next step involves leveraging the developed model for fault warning.

## Figures and Tables

**Figure 1 sensors-23-09762-f001:**
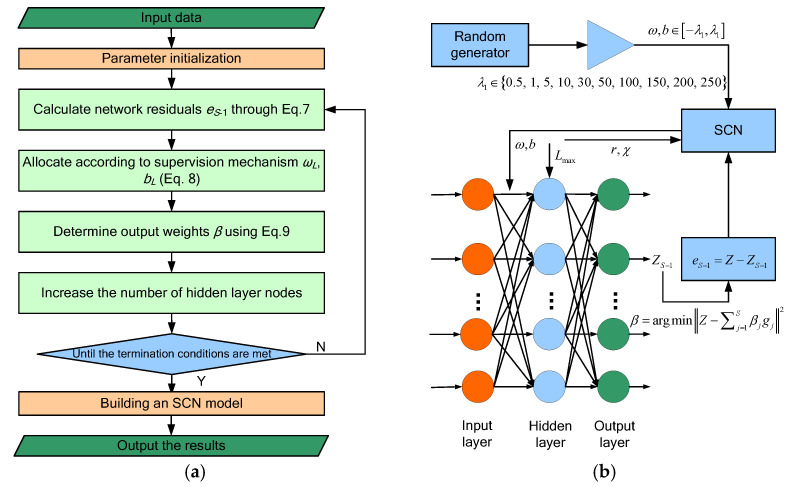
The calculation flow diagram and basic structure diagram of the SCN model. (**a**) The flowchart of the SCN model; (**b**) Schematic diagram of SCN model structure.

**Figure 2 sensors-23-09762-f002:**
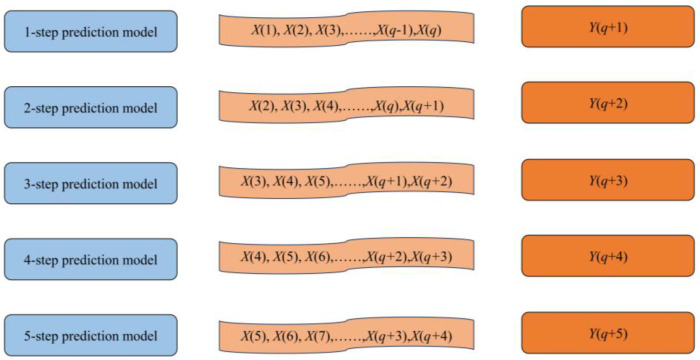
Recursive multistep prediction model.

**Figure 3 sensors-23-09762-f003:**
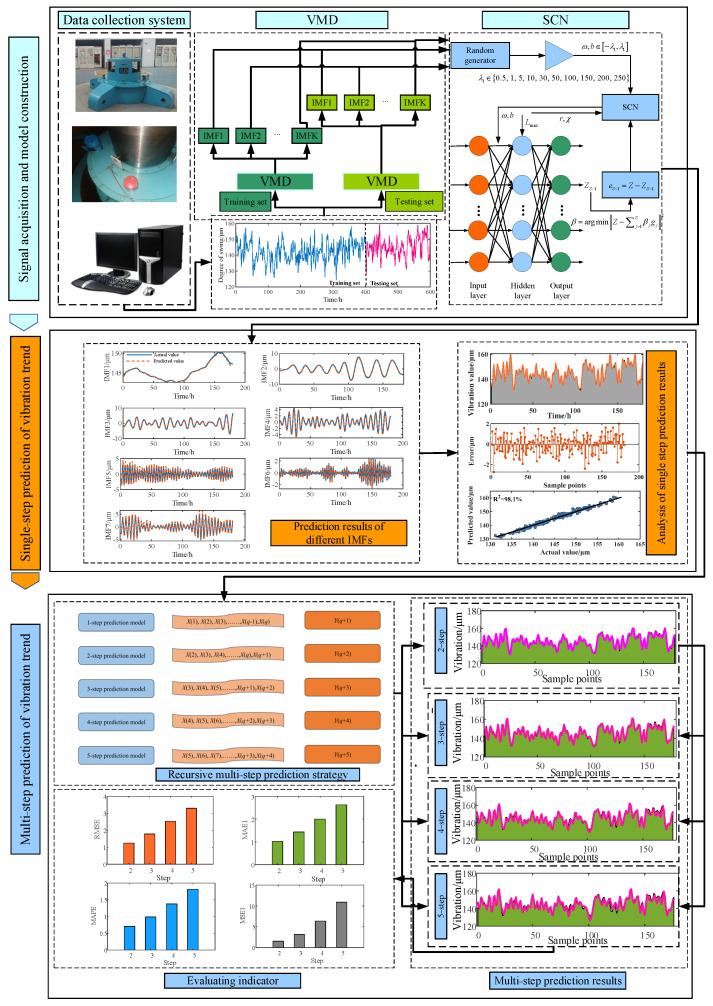
Multistep prediction model for vibration trends of HGUs based on VMD and SCN.

**Figure 4 sensors-23-09762-f004:**
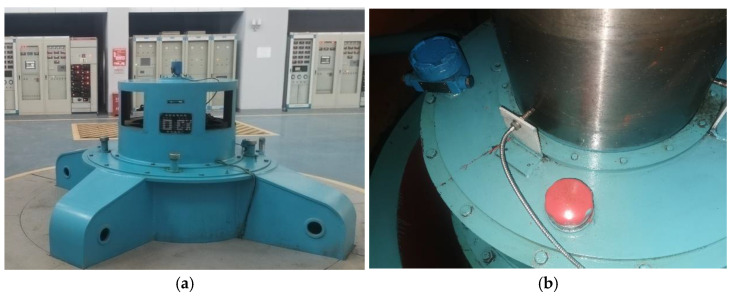
Basic situation of the unit. (**a**) Appearance of HLJ267-LJ-175 hydraulic turbine; (**b**) location of sensor installation.

**Figure 5 sensors-23-09762-f005:**
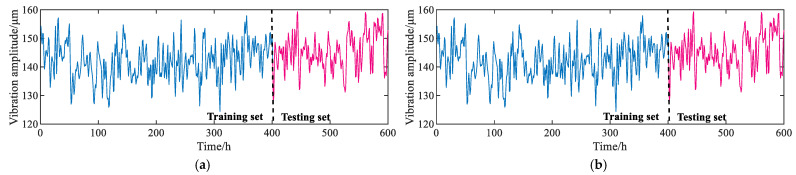
The waveform of the HGU vibration signal. (**a**) *X*-direction vibration waveform of turbine upper frame; (**b**) *X*-direction swing waveform of turbine guide bearing.

**Figure 6 sensors-23-09762-f006:**
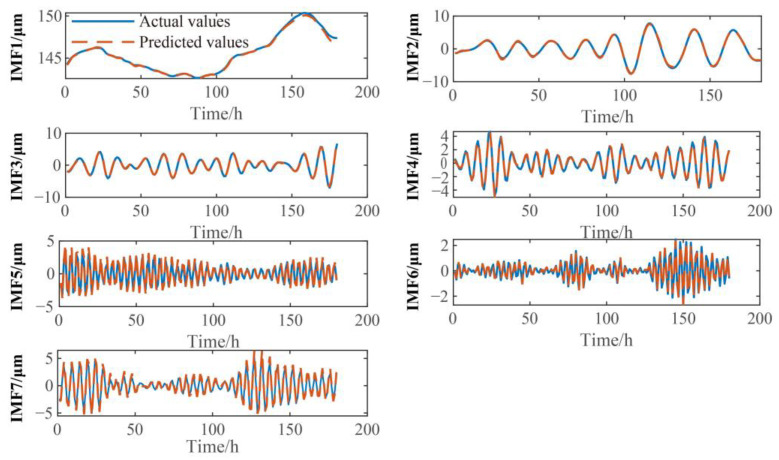
Prediction results for the different IMF components of *X*-direction vibration signal of upper frame.

**Figure 7 sensors-23-09762-f007:**
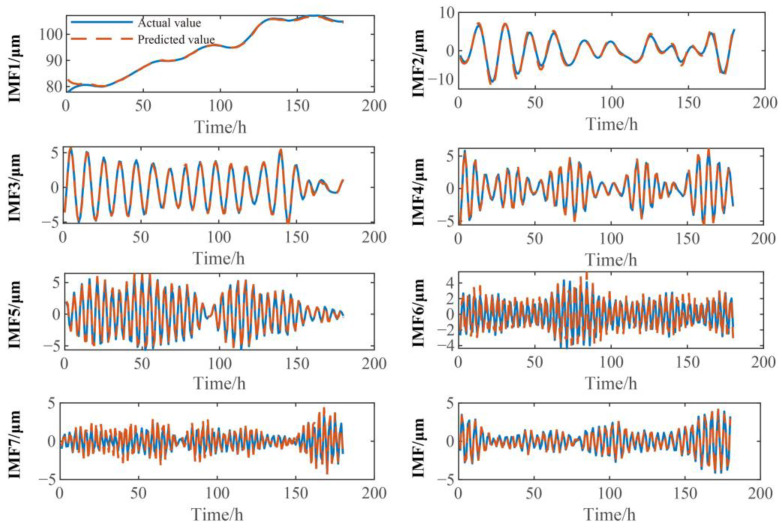
Prediction results for the different IMF components of turbine guide bearing *X*-direction swing signal.

**Figure 8 sensors-23-09762-f008:**
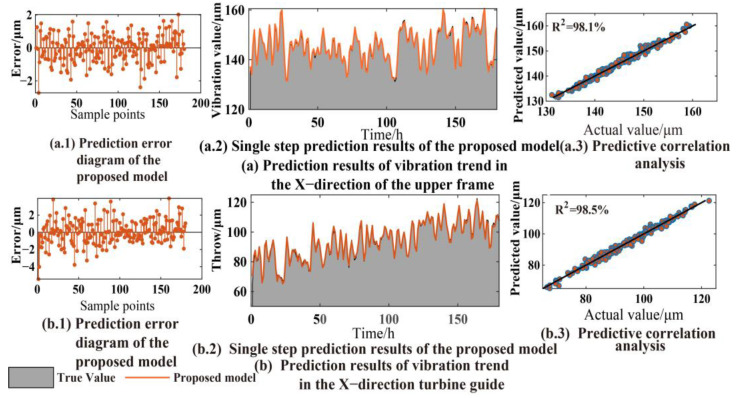
The vibration trend prediction results of the VMD–SCN model.

**Figure 9 sensors-23-09762-f009:**
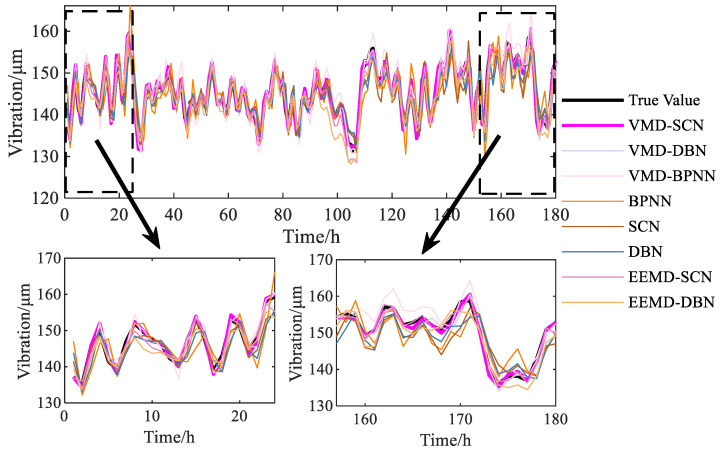
Comparison diagram of different model forecasting (*X*-direction vibration signal of upper frame).

**Figure 10 sensors-23-09762-f010:**
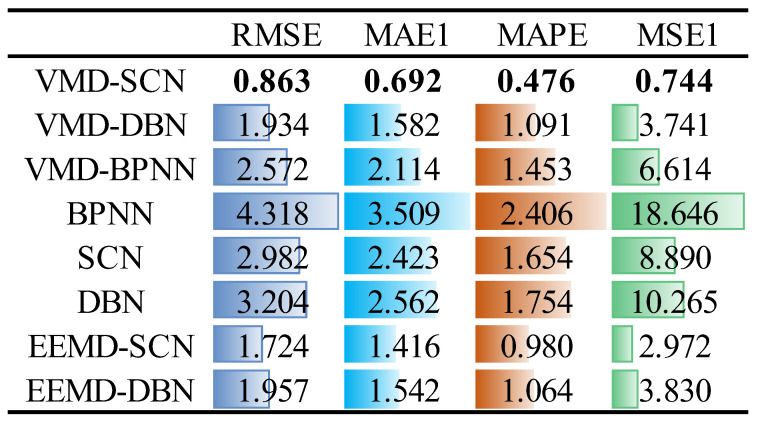
One-step prediction performance indicators of different models (*X*-direction vibration signal of upper frame).

**Figure 11 sensors-23-09762-f011:**
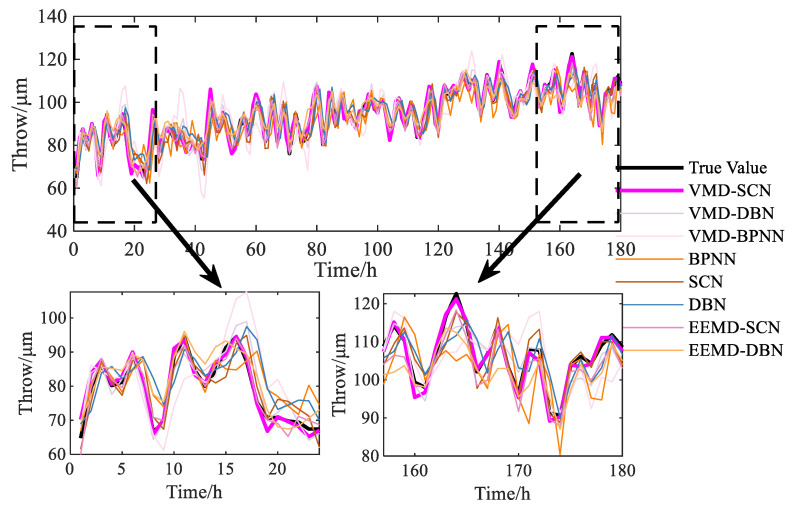
Comparison diagram of different model forecasting (turbine guide bearing *X*-direction swing signal).

**Figure 12 sensors-23-09762-f012:**
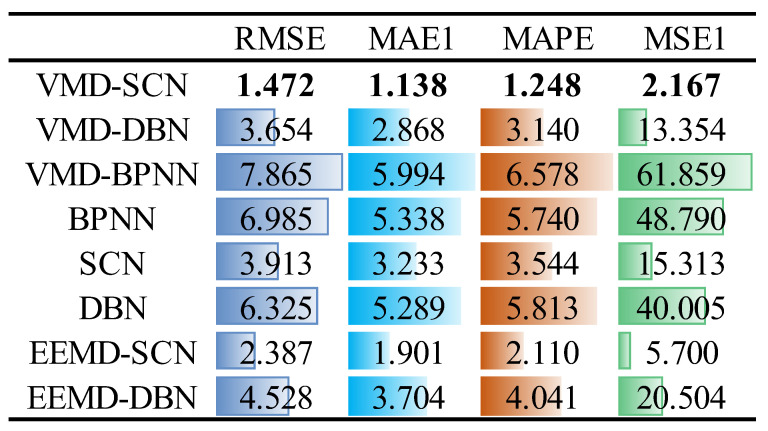
One-step prediction performance indicators of different models (turbine guide bearing *X*-direction swing signal).

**Figure 13 sensors-23-09762-f013:**
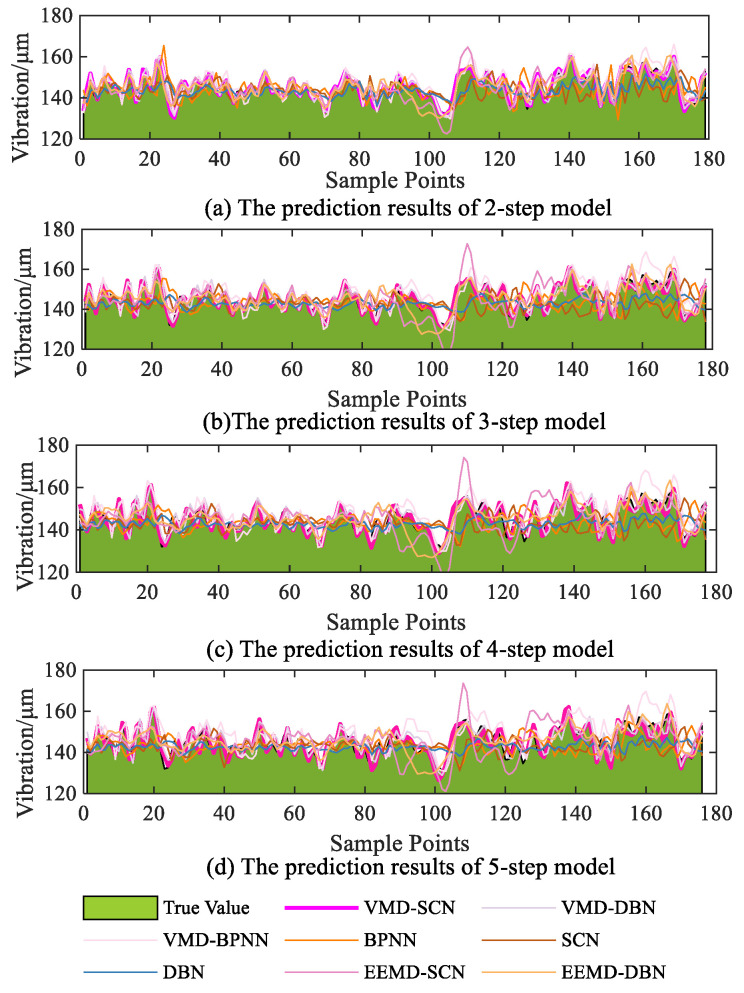
Comparison diagram of different models’ multistep forecasting (X-direction vibration signal of upper frame).

**Figure 14 sensors-23-09762-f014:**
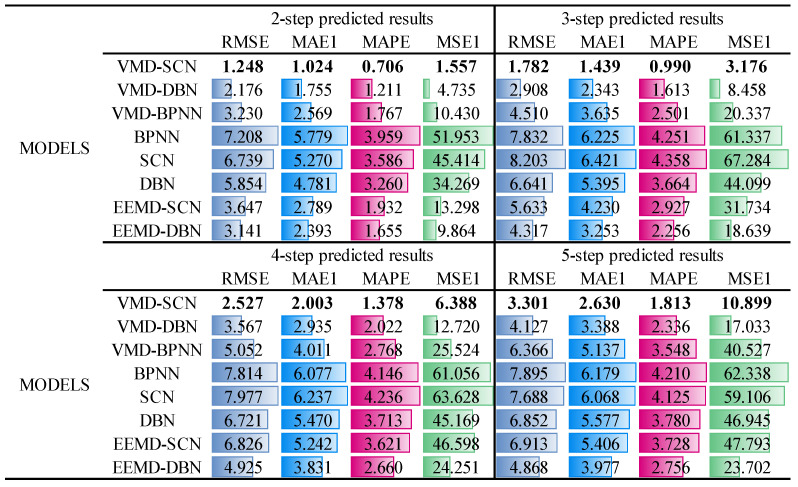
Multistep prediction performance indicators of different models (*X*-direction vibration signal of upper frame).

**Figure 15 sensors-23-09762-f015:**
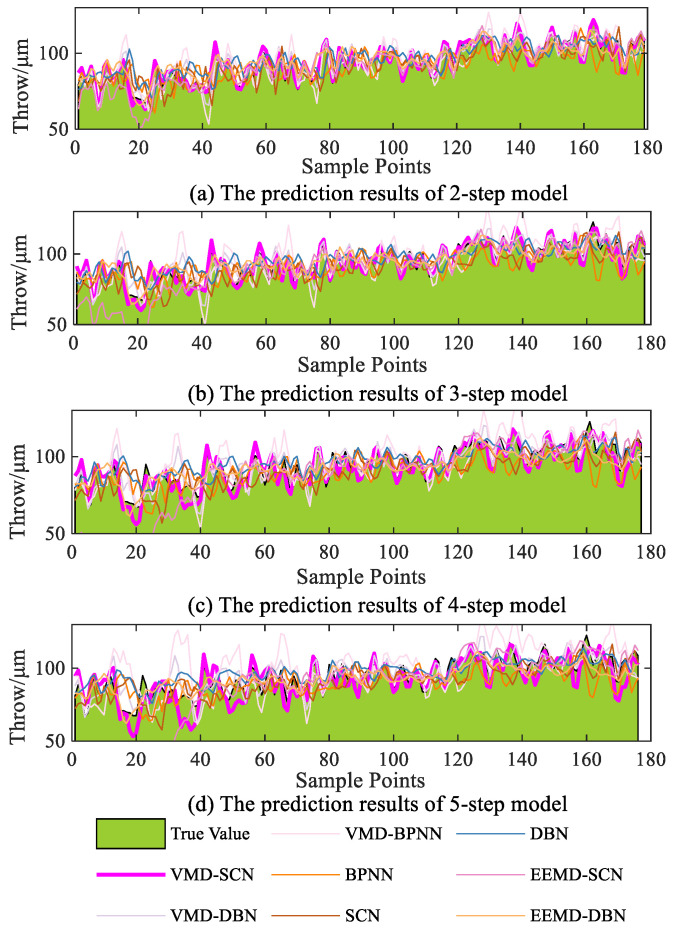
Comparison diagram of different models’ multistep forecasting (turbine guide bearing X-direction swing signal).

**Figure 16 sensors-23-09762-f016:**
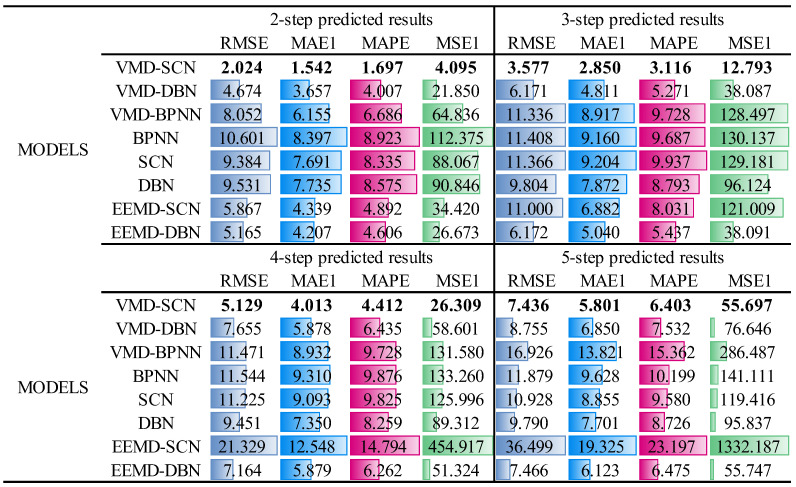
Multistep prediction performance indicators of different models (turbine guide bearing *X*-direction swing signal).

**Table 1 sensors-23-09762-t001:** Normalized center frequencies of IMF components at different decomposition levels.

	IMF1	IMF2	IMF3	IMF4	IMF5	IMF6	IMF7	IMF8	IMF9	IMF10
*K* = 3	4.78 × 10^−6^	0.074	0.243							
*K* = 4	1.71 × 10^−6^	0.067	0.181	0.270						
*K* = 5	6.51 × 10^−7^	0.032	0.078	0.162	0.268					
*K* = 6	7.43 × 10^−7^	0.036	0.088	0.152	0.212	0.269				
*K* = 7	6.06 × 10^−7^	0.031	0.072	0.128	0.170	0.222	0.272			
*K* = 8	5.94 × 10^−7^	0.030	0.070	0.124	0.164	**0.211**	**0.237**	**0.277**		
*K* = 9	5.89 × 10^−7^	0.030	0.070	0.123	0.162	0.197	0.225	0.254	0.285	
*K* = 10	5.87 × 10^−7^	0.030	0.070	0.123	0.162	0.197	0.224	0.255	0.274	0.312

## Data Availability

Data are contained within the article.

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
