# Peer review of "A Multistep Prediction Model for the Vibration Trends of Hydroelectric Generator Units Based on Variational Mode Decomposition and Stochastic Configuration Networks"

_sensors, 2023, doi:10.3390/s23249762_

Round 1

Reviewer 1 Report

Comments and Suggestions for Authors

1. The authors show me very poor understanding on ethics in academic research because they are using SCN to resolve the problem but intentionally ignored the original publication of SCN (D. wang and M. Li, TCYB 2017). Such a behavior cannot be accepted. 

2. The citations 13, 18, 25 are related to a very controversial term ELM, which is identified as being unacceptable  in the community indeed. These citations will result in further misleading to others. the authors need to learn how to do healthy research with high standard ethics concept. See comments on ELM here http://elmorigin.wix.com/originofelm

The submission cannot be accepted for publication before these issues are addressed properly.

Author Response

Please refer to the attachment for reply comments.

Reviewer 2 Report

Comments and Suggestions for Authors

In this work is proposed a combined multi-step prediction model based on variational modal decomposition VMD) and stochastic configuration networks (SCN).

Although the proposal is interesting, there are some issues that must be addressed.

1. The novelty and contribution have to be emphasized in the abstract and in the introduction sections.

2. What are the differences of the proposed IMF estimation in comparison with other implementations, please include a brief discussion by considering the following works: https://doi.org/10.1109/TIA.2016.2637307 ; https://doi.org/10.1016/j.measurement.2020.107660

3.  Please consider to include more details in the description of section 3. Figure 3 seems to show all the details but the corresponding description doesn't give details.

4. The conclusion should be written as "conclusion". In the current version, conclusion are given as a summary of the steps performed to accomplish the proposal.

5. What are the disadvantages or limitations of the proposed work, can the authors include a brief discussion about that in the conclusion section.

Comments on the Quality of English Language

Please make a revision of the whole document in order to avoid mistakes.

Author Response

(The authors gave the same response as above.)

Reviewer 3 Report

Comments and Suggestions for Authors

This paper proposes a multi-step prediction model for the vibration trend of hydroelectric generators (HGU). The model uses variational modal decomposition (VMD) and stochastic configuration networks (SCN). The applicability of the methods is validated experimentally. This is a well-written practically-oriented paper and I have only a couple of comments:

1. What was the reason for choosing VMD and SCN for this purpose? There is a comparison in Section 4.3 with other approaches, but the authors could comment their suitability/benefits already in Introduction.

2. I would integrate the short Section 3 as a part of Section 2.

3. The concluding section needs few words about the future research needs.

Author Response

(The authors gave the same response as above.)

Round 2

Reviewer 1 Report

Comments and Suggestions for Authors

This revised version reads better with additional information and references. However, there are still some serious issues to be further fixed. My detailed comments and suggestions are given below:

1. SCN refers to stochastic configuration networks, rather than stochastic configuration neural networks. Please go through the manuscript and correct this mistake. 

2. I noticed that the authors made comparisons with the so-called ELM in simulation section. This is misleading because ELM is simply a renamed term of RVFL nets, which was stolen by Huang GB et al and kept promoting such an abused term  in China through making conferences and special issues in Neurocomputing , resulting in thousands of junk publications. The authors must read the following link to understand how sick the so-called ELM is.

http://elmorigin.wix.com/originofelm

To avoid further misleading in the community, all associated ELM references must be removed from the list of references. This is compulsory for publications with high standard of ethics.

Comments on the Quality of English Language

English writing is acceptable in general. However, there are many spaces to make the sentences more informative.

Author Response

(The authors gave the same response as above.)
